# Digital Violence in University Student Couples: England vs. Spain

**DOI:** 10.3390/ijerph21070926

**Published:** 2024-07-16

**Authors:** Ángel Hernando-Gómez, Delia Montero-Fernández, Antonio Daniel García-Rojas, Francisco Javier Del Río Olvera

**Affiliations:** 1Department of Social, Developmental and Educational Psychology, University of Huelva, 21071 Huelva, Spain; angel.hernando@dpsi.uhu.es; 2Metropolitan School District of Lawrence Township, Indianapolis, IN 46220, USA; delia.monterofdz@gmail.com; 3Department of Pedagogy, University of Huelva, 21071 Huelva, Spain; 4Department of Psychology, Institute for Biomedical Research and Innovation of Cadiz (INiBICA), University of Cádiz, 11510 Puerto Real, Spain; franciscojavier.delrio@uca.es

**Keywords:** digital violence, cyberstalking, university students, social networks, information and communication technology

## Abstract

This research studies the prevalence of digital violence exercised through new information and communication technology (ICT) among university couples. A comparative study was carried out in England, United Kingdom, and in Spain with 831 participants. A quantitative methodology was applied with different sampling: in the United Kingdom, 303 (*M*_Age_ = 22.79; SD; 47.32; 58.7% male) and in Spain, 528 (*M*_Age_ = 24.29; SD = 21.41; 69.5% female). An ad hoc questionnaire was used, created for the detection, measurement and analysis of digital violence within affective-sexual relationships. The results reveal proportions of 51.04% and 49.82% in the perception of digital violence through electronic devices in dating relationships among young people; 15.84% and 11.05% in the prevalence of digital violence in young couples’ relationships; 9.36% and 6.17% in the prevalence of traditional violence; and 35.78% and 22.43% in the tolerance of digital violence among students, for the English and Spanish samples, respectively. The results also show a slightly lower prevalence of digital violence in the Spanish sample with respect to the English sample, where females scored slightly higher in the perpetration of digital violence. There is a need to develop awareness, training and prevention programs against digital violence in the university context.

## 1. Introduction

Recalling the words of the philosopher Byung-Chul Han, “today we do not torture, but we “post” and “tweet””, [1] we begin the presentation of a new form of violence born in the 21st century. A violence exercised through the use of information and communication technologies (ICT), that arises given the consequent digitalisation of socialisation and interpersonal relationships among the young population [2], especially in the world of education in universities with remote teaching practices [3]. According to data from the European Statistical Office (Eurostat), ICT immersion in the population is estimated at 98% in Spain in 2020 and 96% in the United Kingdom in 2019.

Derived from the new risks arising from the immersion in ICT in society, the university population and their affective-sexual relationships have become the target of the deployment of this digital violence due to the fact that university students represent the main recipients of ICTs and that, because of their age, are a digitally native population. Referring to national and international scientific evidence, several studies [4] have found the need to study the impact of ICT on the lives of university students due to the problematic use of ICT and their social networks. This is because the behaviours of control or surveillance of the partner or ex-partner in digital spaces form one of the most common expressions of digital violence [5]. Other research has recorded manifestations of this violence in the form of sending threatening or abusive messages and emails [6,7], posting humiliating photographs [6,8], as well as monitoring and surveillance of a partner [8].

Likewise, in terms of the prevalence of digital violence, one of the studies reviewed found that written aggression, in the form of insults/offences or threats, reached 70%, making it the most common type of violent action within the university context in Spain among the profiles of cyber-perpetrators of this violence [7]. Another recent study, conducted at the University of Granada (Spain), found that 6.2% of participants experienced cyberstalking and digital violence, 6.7% harassed others, 8.4% were victims/offenders and 68.8% had no role. Compared with the university context in the UK, research at several universities in Manchester (England) found that between 20% and 34% of participants had reported experiences of cyberbullying [9]. Consistent with other research conducted in the same country among non-students, an online survey of 324 participants, 34% of whom were male, reported 92% as having experienced digital violence and cyberstalking [10]. Similarly, one of study reports that 20.5% of its participants identified themselves as victims of cyberstalking [11]. Other authors have found that 46% of their respondents had been victims of cyberstalking [12], while another group of researchers indicated that 35% of respondents had been victims of technology-facilitated stalking [13]. Thus, given the numerous research studies and all of their findings, the emerging social problem among the university population is raised [3,14].

Furthermore, this brief scientific review reveals discrepancies in the way in which the phenomenon in question has been operationalised, with different terminologies and meanings. Therefore, the lack of consensus in the results obtained due to this ambiguity is striking [15,16]. Most studies refer to cyberstalking [7,11]. However, in this paper, the concept is broadened to include digital violence, which goes beyond cyberstalking, encompassing coercive control as one of the indicators, in the form of repeated efforts at harmful and unjustified control [17]. At the same time, this digital violence encompasses intimidation, domination, threat, and surveillance through the use of digital media [18].

Of the risk factors for digital violence in its cyber-perpetration and cyber-victimisation, the most commonly found have been age, maturity and stability of the affective-sexual relationship, influencing the decrease in aggression and abuse through electronic media [19]. Additionally, distance in couple relationships has been presented as a factor in the increased use of social networks to monitor the partner [20], as jealousy towards the other partner has been shown to lead to controlling behaviours [21]. Similarly, gender has been extensively studied in the production of digital violence. Some studies show that women perpetrate more digital violence compared with men [11,22,23]. However, other studies claim that men are more likely to perpetrate these assaults online [24]. Meanwhile, other researchers have found no significant differences between the gender in the perpetration of online aggression and abuse towards their partners [9,25,26]. Therefore, in this case, there is no uniqueness in the results and the sex of the participants cannot be deduced as a risk factor or variable to be taken into account. Thus, this research proposes the hypothesis that the digital violence detected will form a pattern of bidirectional or mutual violence in the relationships of university couples.

On the other hand, there is evidence of a legitimisation and normalisation of digital violence in affective-sexual relationships by the young population. The affirmation of the perpetration of abusive behaviours towards partners has been demonstrated, as has the lack of consideration of abusive actions by young participants [27]. As a consequence, the tolerance, legitimisation and normalisation of digital violence behaviours can lead to its manifestation within the relational dynamics of a young couple [28].

This literature review shows that awareness-raising among young people is crucial to the identification of this social problem [29]. Although previous research on digital violence in the university context has been conducted in the UK and Spain, there is a scarcity of specific studies on digital violence in both countries, perhaps due to the aforementioned ambiguity in its conceptualisation. Therefore, there is a need for more research on digital violence in the context of university students. This work is carried out in different university contexts: in the United Kingdom, specifically in the northwest of England, and in the southwest of Spain. On the one hand, there is a certain similarity between the countries in terms of their political and economic situation, safeguarding cultural and contextual differences, while, on the other, there has been greater social awareness-raising work in recent years in Spain, especially within the student movement and its university context. For this reason, the main objective of this research is to analyse the digital violence that occurs through the use of ICT and all commonly used electronic devices in affective-sexual relationships in the university population. The analysis of the university context provides a more easily comparable sample population in terms of the profile of the participants as students. Therefore, the specific objective of this work is to detect and measure the prevalence of digital violence among university couples, comparing the populations of Spain and England, in order to pursue its prevention.

## 2. Materials and Methods

The methodology proposed for this work is framed within the framework of explanatory research methods. This research proposes a quantitative methodology, with an ex post facto research design [30]. In the first study, carried out in the United Kingdom, a non-probabilistic purposive or discretionary sampling was carried out, with the use of an ad hoc questionnaire, for the detection, measurement and analysis of digital violence within affective-sexual relationships. In the second study, conducted in Spain, the sampling was two-stage random cluster sampling with the use of the same ad hoc questionnaire. Data collection was carried out during the 2021/2022 school year, with the first semester of fieldwork taking place in England (UK) and the second semester in Spain.

### 2.1. Participants

The English study was conducted at the University of Central Lancashire (UCLan) in the English city of Preston. Using the total university population of 38,000 students as the population data, with a confidence level of 95%, and assuming a sampling error of 5.61%, a sample of 303 students was obtained. Of the 303 persons in the sample, 41.3% were female (125) and 58.7% male (178). The mean age was 22.79 years. The Spanish study was carried out at the University of Huelva (Spain). The student population of this university amounts to about 11,000 students. Using the total student body as the population data, with a confidence level of 95% and assuming a sampling error of 4.17%, a sample of 528 students was obtained. Of this total number, 69.5% were female (367) and 30.5% were male (160). The mean age was 24.29 years, with a standard deviation of 21.41 years.

### 2.2. Instrument

Due to the lack of a complete and adequate instrument for the analysis of violence through electronic media in affective-sexual relationships, at national and international level, the construction of an ad hoc questionnaire was proposed, which was used for the study in the United Kingdom. The first part of this ad hoc instrument is composed of sociodemographic variables and relational variables in the dating relationship that preserve the anonymity of the questionnaire. Among these variables are the number of relationships and their duration, sexual orientation, type of relationship, among others. These socio-demographic and relational data in the dating relationship were collected through multiple choice and closed questions. The second part is composed of five blocks with a total of 90 items. Block one, composed of 12 items, assesses the perception of violence and strategies of digital violence; block two, composed of 27 items, measures the prevalence of new forms of digital violence in dating relationships; block three, with 26 items, compares the prevalence of screen and non-screen violence; block four, consisting of 13 items, assesses the tolerance of this violence through ICT; block four, consisting of 13 items, assesses the tolerance of digital violence among young people; block five, with the last 12 items, explores possible causes and consequences of new forms of digital violence in relationships. In this second part, a Likert-type response format was chosen with a response option on the aggressions suffered and perpetrated, with a range of values of 1 “disagree”, 2 “slightly disagree”, 3 “slightly agree”, and 4 “agree” for block one; and a range of values of “never”, “seldom”, “sometimes”, and “often” for blocks two, three and four of the instrument. “Often” means that it has occurred six or more times in the relationship; “sometimes”, about three or four times; and “seldom”, once or twice in the relationship.

### 2.3. Procedure

In both studies, the inclusion criteria were age of majority, belonging to the university as a student and having or having had a dating relationship. As exclusion criteria, participants with a dating relationship of less than three months were not admitted. The questionnaires were administered in the presence of the main researcher, insisting on anonymity and sincerity in the answers. The time taken to complete the questionnaires was approximately 15 min.

Regarding data analysis, Cronbach’s alpha was calculated for the analysis of the reliability of the questionnaire, as well as the item–total correlation for the analysis of the items. Data analysis was carried out using statistical and data management software tools such as, in this case, SPSS 24.0.

## 3. Results

In comparing the results of the studies carried out in the United Kingdom and Spain, it is worth mentioning, firstly, the total number of participants in the different samples, given that the English sample has 303 participants while the Spanish sample has 528 people. The average age in the English study is 22.79 years and 24.29 years in the Spanish study. At the same time, due to the different data collection techniques used in both field studies, there is a greater variety in the academic disciplines being studied by students at the English university than at the Spanish university. In terms of sexual orientation, the representation of homosexuality, bisexuality and others is very low in both samples, with heterosexuality remaining predominant at 82.2% (93 females and 156 males) in the English sample and 93% (347 females and 142 males) in the Spanish sample. Regarding marital status, mainly due to the average age of the samples, among other reasons, the vast majority designated being single, 69.2% in England and 96% in Spain. In relation to the type of family in which they had grown up, the nuclear family stood out by a large absolute majority in the different samples, with 63.4% in the English sample and 82% in the Spanish sample, followed by 20.8% in a single-parent family in the first study and 12% in the second. Regarding the socio-economic status of the family for in Spain and England, the majority of respondents in both samples said they were in the middle socio-economic status, while a respective 48.6% and 47.3% said they were in the upper-middle range, and 45.1% and 49.8% said they were in the lower-middle. Looking at the educational level of the family, both studies again show very similar results. The mother figure obtained a majority of 67.8% with a “General Certificate of Secondary Education (GCSE)” in the English study, while in the Spanish study the percentage was 59.9%, both of which were followed by the option of “Educated up to Primary Education level”. The parent figure was 74.6% and 18.2%, respectively, for “General Certificate of Secondary Education (GCSE)” and “Certificate of Higher Education or more” in the English questionnaire and 78.3% and 14.7% for “General Certificate of Secondary Education (GCSE)” and “Certificate of Higher Education or more” in the Spanish questionnaire.

Looking at the duration of relationships, it can be seen that, in the English sample, there are more relationships of one to two years (34.4%), followed by relationships of three to five years (28.2%), as is the case in the Spanish sample, with 20.4% and 22%, respectively. However, in the Spanish relationships, the same percentage is also seen in the five-to-ten-year duration (22.3%), so that slightly longer lasting relationships are seen in the Spanish sample. As for the type of relationship that the participants claim to have, there is more variety in the English sample where, firstly, the serious and/or stable relationship (46.8%) and, secondly, the casual relationship scores higher (33.2%). In the Spanish sample, it is the serious and/or stable relationship that is obtained with a large majority (71.6%). Of all these relationships and in terms of the frequency of direct and face-to-face contact, 19.3% maintain direct and face-to-face contact with the partner more than once a day among English couples, while this same percentage in the Spanish sample corresponds to 12.7%. The highest percentage for the English sample is 36.3% for every day, followed by 26.1% for seeing each other two or three times a week. The highest percentage for the Spanish sample in this respect is 42.8% for seeing each other two or three times a week, followed by 28.6% for seeing each other every day. What can be deduced is that English couples have a higher frequency of direct, face-to-face contact than the Spanish couples surveyed.

Of the total results obtained in the blocks, excluding block five which looked at risk factors and possible consequences, the UK study showed 51.04% in the perception of digital violence through electronic devices in dating relationships among young people, 15.84% in the prevalence of digital violence in young couples’ relationships, 9.36% in the prevalence of traditional violence, and 35.78% in the tolerance of digital violence among young people. The Spanish study revealed 49.82% in the perception of digital violence through electronic devices in dating relationships among young people, 11.05% in the prevalence of digital violence in young couples’ relationships, 6.17% in the prevalence of traditional violence, and 22.43% in the tolerance of digital violence among young people.

Beginning with the comparison of block one of the ad hoc questionnaire, which studies the perception of violence and strategies of control and abuse through electronic devices in dating relationships between young people, the majority of responses in this block pointed to a slightly lower perception and awareness of digital violence in the English sample compared with the Spanish sample. A majority of responses in this block pointed to a slightly lower perception and awareness of digital violence in the English sample compared with the Spanish sample, except for some significant ones, such as item 6, below. For example, in item 1, which asks about reading the partner’s conversations on social networks, the English sample showed 31% “agree” and 30.4% “somewhat agree”. In the Spanish sample, 20% said they “agreed” and 33.5% said they “slightly agree”. Thus, the Spanish sample was divided in its opinion on this issue, while the English sample showed a slight acceptance and agreement with this action. Similarly, in item 2, which asks about knowing the partner’s passwords in virtual spaces, the English sample answered 20.8% “agree” and 30% “slightly agree”, while the Spanish sample showed 12.6% “agree” and 27.8% “slightly agree”. This again demonstrates the subtle difference between the samples in relation to the perception of the above-mentioned action. It is interesting to note item 6 in Figure 1, which asks about looking at a partner’s daily internet connections, where the overwhelming disagreement of the English sample, compared with the Spanish sample, can be seen.

Likewise, it is worth highlighting the difference in the perception that participants show for item 7 on the continuous control of the partner in social networks. Figure 2 below clearly shows how the English sample expressed greater agreement with this type of behaviour than the Spanish sample. Likewise, in both samples, the upward trend towards online control and abuse became clearer as the degree of engagement in their own relationships increased.

Continuing with the comparison of block two, which measures the prevalence of new forms of digital violence in young couples’ relationships of both partners. The overall low, but significant, prevalence of digital violence in young couples’ relationships, according to respondents in the UK and Spain, should be highlighted. In the responses to all of the items in this block, there is a tendency for respondents to indicate that the partners of the participants in all cases showed a higher frequency of the behaviours framed as digital violence. However, given the overwhelmingly negative responses in the Spanish sample to this type of online abuse and control, it is worth noting the “slight tolerance” of the English sample to the same actions. This “slight tolerance” is reflected in the higher percentages of “sometimes” and “often” responses for each item. As an example, it is worth highlighting the comparison with item 13 in both samples, which asks with whom the subject interacts with on social networks; here, the percentage of the response “often” is 22.5% in the case of the partners of the surveyed participants and 14.4% for the prevalence of their own behaviour for this same action in England. In Spain, the percentages for the same responses are much lower, with 7.7% in the case of the partners and 2.2% in the case of the participant themself. Figure 3 shows the results, according to what the participants think about the behaviour of their partners and themselves, showing a clear difference between the two samples. The tendency towards horizontality in the English sample indicates a greater affirmation in the production of the action in the English sample than in the Spanish sample, for all of the options.

In item 15, on looking at the movements of the partner in virtual spaces, in the UK the percentage of the response “sometimes” was again found to be 27% for the partners of the participants and 18.5% for themselves; in Spain, the percentages again lowered for the same responses, representing 15% in the case of the partners and 10% in the case of the participants themselves. Item 34, about sending naked pictures to partners, showed, in the English sample, 16.7% reporting “often” and 17.9% “sometimes” for boys’ partners; and 13.2% “often” and 16.5% “sometimes” for girls’ partners. The results for the participants themselves, as opposed to their partners, were only 1% higher for both options, which suggests that slightly more of the surveyed participants have engaged in this behaviour than their own partners, as reflected in Figure 4 below. In the Spanish study more boys (5.8%) revealed sending naked pictures to their partners “often” than girls (2.5%); however, more boys (5.7%) indicated that their partners engaged in this behaviour “often”, compared with 3% of girls’ partners.

Referring to item 36, about posting erotic photos of their partners in virtual spaces after an argument, in the English sample no boys admitted that their partners had done this “often” and 3.3% of girls stated that their partners had done it. Considering that the vast majority of the sample is heterosexual, it follows that no girls in a couple position had posted erotic photos of their partners after an argument, when this action was carried out by some of the girls’ partners. Accordingly, the first-person responses for this item corresponded with the previous ones, as no girl admitted to having done this “often” and a percentage of 2.6% of boys declared they had. In the Spanish sample, only one girl (0.3%) confirmed, in the first person, having posted erotic photos of her partner on the internet after an argument. When referring to the partner’s actions in the same sample, one boy (0.6%) and one girl (0.3%) acknowledged that their partners always do it. On the other hand, the remaining items in this second block showed a prevalence of digital violence that did not reach 5% for “often” or 10% for “sometimes”, either for boys or for girls and either in the first person or when referring to their partners.

In block three, the prevalence of the use of violence off-screen will be studied, that is, the gender-based violence that can occur in affective-sexual relationships will be assessed by considering different possible situations. In block three, an almost negligible prevalence of the use of off-screen violence is detected in both samples in many of the responses analysed. The majority of items are denied by almost the entire sample population of this research, reaching 90% in the case of the United Kingdom and almost 100% in Spain for both the responses and when referring to the partner or to the participant him/herself. By way of example, item 48 studies “isolating the partner from family and friends” and is analysed from the individual’s own perspective of the partner’s actions. In the English sample, 86.6% report “never”, 6.3% “seldom”, 4.9% “sometimes”, and 2.1% “often”. In this sample, no girl admitted to isolating her partner from her family and friends. In the Spanish sample, with the same options regarding partners, 90% were marked as “never”, 5.6% as “seldom”, 2.3% as “sometimes”, and 1.9% as “often”. Some 2.5% of girls admitted that their partners did this “often” compared with 0.6% of boys. Likewise, no boys admitted to having done this “often”, in the first person. There is a slightly higher prevalence in the English sample than in the Spanish sample, and it is worth noting the absence of responses in the aforementioned options from both sexes. At the same time, item 63, which regards “insulting, pushing or hitting one’s partner in an argument or fight”, observes the findings of the partners of the individuals surveyed. In the English sample, when referring to the actions of the participants’ partners, 84.9% were found to be “never”, 8.2% “seldom”, 4.8% “sometimes”, and 2.1% “often”. This final 2.1% is striking in view of the normality of these actions in the affective-sexual relationship in the English study, where 2.3% resulted from male partners and 1.7% from female partners. Going deeper, it was found that no girl admitted to doing this in the first person and only two boys (1.2%) did it “often”. In the Spanish sample, we again find a more negative percentage towards these behaviours, with 90% being marked as “never”, 8% as “seldom”, 1.2% as “sometimes” and 0% as “often”. In other words, the comparison of block three for the English and Spanish sample highlights the similarity in the tendency to deny the existence of explicit gender-based violence within their affective-sexual relationships. However, the Spanish sample more absolutely denies these behaviours in all cases.

In block four, the comparison between the two samples is very striking. This block studies the tolerance of this violence through new technologies in young people, i.e., how the participant accepts and carries out behaviours in which this type of online abuse is denoted and shows differences in terms of tolerance between the United Kingdom and Spain. While in the English sample the percentages for item 66 are 25% and 31% who admit that they “often” and “sometimes” tell their partner what they are doing and where they are at all times, respectively, in the Spanish sample it is 20% and 41% for the same options. In this case, the parity in the results of the two samples for the same item is striking, and it is also worth noting the slightly higher tolerance of respondents in Spain towards telling their partner about all online activity. Looking at item 71, the results for the Spanish sample are that 17.7% of respondents do it “often”, 12% do it “sometimes”, 23% say they do it “seldom” and 47.3% say they do it “never”. For the English sample the results are around 30% for “never” and “often”, and around 20% for “seldom” and “sometimes”. The comparison of this item for both samples is presented in the following graph in Figure 5.

It is also worth mentioning item 78, on the permissibility of insults or threats through messages in a moment of discussion, as differences are observed in both populations. While in the English sample the percentages are 14.8% for “often”, 12% for “sometimes”, 21% for “seldom” and 51.5% for “never”; in the Spanish sample these same categories correspond to values of 1.3%, 5.8%, 12.3% and 80.6%, respectively. Again, there is a significant trend in the English sample towards greater permissiveness in abusive online behaviour compared with the Spanish sample.

For block five, risk factors and possible consequences of digital violence in the affective-sexual relationship are studied. Reviewing the results of the studies in England (UK) and Spain, the three items, 85, 86 and 87, on different situations of violence that could cause the break-up of the affective-sexual relationship are worth highlighting. In both samples, the responses are very similar; however, once again, the Spanish sample shows a larger response when compared with the English sample, both for the “agreement” to break up the relationship when there is a push in the middle of an argument (item 85) and when there is physical violence as such (item 86). At the same time, it is curious how both samples show a broader agreement when it comes to not tolerating physical violence in their affective-sexual relationships, but they do not seem to cross out a push as an example or behaviour of physical violence, as presented in Figure 6 below.

Item 87 asks whether “I would not leave my partner if he/she controlled me through electronic media”. In this case, it should be noted that the English sample shows a slightly higher and significant opposition (32.3% disagreement) towards agreeing to break the relationship in the case of this type of behaviour. However, the percentage of “agreement” is far from reaching the majority in both samples: 17.2% “agree” and 22.7% “slightly agree” in the English sample, while 8.1% “agree” and 12% “slightly agree” in the Spanish sample. Therefore, the lack of identification of these types of attitudes and behaviours as online violence is interpretable. Finally, it should be noted that all of the “agree” responses to the items in this block were scored slightly higher by boys than girls in the English sample.

Finally, item analysis and scale reliability were estimated using Cronbach’s alpha coefficient, with a value of 0.955 in the English study and 0.945 in the Spanish study, determining a high internal consistency in the ad hoc questionnaire. It is also worth mentioning that the sample distribution did not meet the assumption of normality (*p* = 0.000), calculated through the Kolmogorov–Smirnov test, so non-parametric statistics were used. In addition, KMO (0.872) and Bartlett’s sphericity (1.4197.470; *p* < 0.000) tests were performed to verify the appropriateness of conducting the factor analysis.

## 4. Discussion

Reviewing the previous results and the aim of this research study on the prevalence of digital violence among university students and their affective-sexual relationships, the prevalence of digital violence in England (UK) and Spanish studies is confirmed, with low values for each of the factors. However, it should be noted that the prevalence was slightly higher in the English sample, which was composed of more multicultural students when compared with the Spanish sample. The observed prevalence of digital violence in the Spanish study is consistent with that in the work of Strawhun et al. (2013) [9] while in the English study, it is closer to the findings of Berry and Brainbridge (2017) [10]. Nevertheless, our results are below those found in some reviewed studies [11,12]. Thus, it is observed that the tendency towards a negative response of disagreement to these assumptions, which are related to the online control and abuse behaviours that young people show towards their partners in affective-sexual relationships, could be explained by a phenomenon of social desirability. Furthermore, it should be remembered that the low percentages in this research could be explained by the non-identification and normalisation of digital violence that has also been detected in previous studies [26,27]. Likewise, and similar to the results of these studies, the prevalence of physical and sexual violence that is displayed off-screen was found to be lower than that of digital violence in both populations.

The differentiation in the identification of perpetration and victimisation stands out, as it is difficult to identify victimisation versus perpetration, especially in the first person. In almost all cases, the participants in this study recognised more perpetration of digital violence in the attitudes and actions of their partners than in their own actions. Furthermore, in many cases there was a negative tendency to perpetrate digital violence as the participants increased the commitment of their relationships, depending on the type of affective-sexual relationship [18]. On the other hand, taking into account the frequency of contact [19], digital violence was more evident in affective-sexual relationships as the frequency of contact was lower between its members, correlating again the opinions of the English students with those of the Spanish students. With regard to the sex of the participants, we allude to the hypothesis put forward and conclude that the abuses and coercive controls of digital violence make up a pattern of bidirectional or mutual violence in the affective-sexual relationships of the English sample [24,25]. In the Spanish sample, many girls scored higher in percentages on perception, tolerance and prevalence of digital violence, consistent with the results of several research papers reviewed above [9,21,22].

Analysing the level of perception and tolerance of digital violence among the participants in the samples, we detect a low perception of digital violence compared with off-screen violence in both populations, with the English sample standing out above the Spanish sample. However, differences are established between the studies on the tolerance and normalisation of the behaviour of pushing, as physical aggression, which scores higher in boys in the English sample. The results show that digital violence can be perpetrated without the presence of physical violence. However, cases in which physically and sexually violent behaviour is detected are also likely to show abuse, harassment and control through electronic devices.

As proposals for future research, it is recommended to continue efforts to overcome the lack of consensus regarding prevalence in the study of digital violence in the scientific literature. It is also advisable to explore the exploration of risk factors in cyber-perpetration and cyber-victimisation of this phenomenon of digital violence in order to act more effectively in its prevention and detection. Finally, in terms of the limitations found, the English sample had 303 participants while the Spanish sample had 528 people. Among the socio-demographic variables, there is also a greater diversity in the nationalities found among the students of the English university UCLan than in the Spanish university of Huelva. Thus, there is a need for a more in-depth analysis with a representative and randomly selected sample, in order to eliminate possible sampling biases. The access and selection of the university populations, in the two-stage random cluster sampling and the incidental sampling, could have biased the sample by university discipline in a less equitable way. Likewise, one could deduce an influence of the university context in which the fieldwork was carried out, related to a tendency towards social desirability and a concomitantly less accurate response to the actions of the participating participants.

## 5. Conclusions

In conclusion, this research highlights the importance and seriousness of the violence that still occurs in affective-sexual relationships. Digital violence is currently recognised as a pressing social problem, leading to social progress in the study and design of various intervention and prevention programmes on the phenomenon in the educational sphere. Among the main problems to combat would be the normalisation of violence in couples among boys and girls, who, although they are able to identify gender violence in adult couples, are not aware that it can also happen at their age and in a non-physical way, in virtual spaces. In this case, the low perception of digital violence among university students in England (UK) compared with Spanish students is highlighted, as extrapolated from the results previously analysed. However, schools and universities in both countries should guarantee that attempts at prevention will encompass the entire population, as attendance in those institutions is compulsory for everyone up to the age of sixteen.

## Figures and Tables

**Figure 1 ijerph-21-00926-f001:**
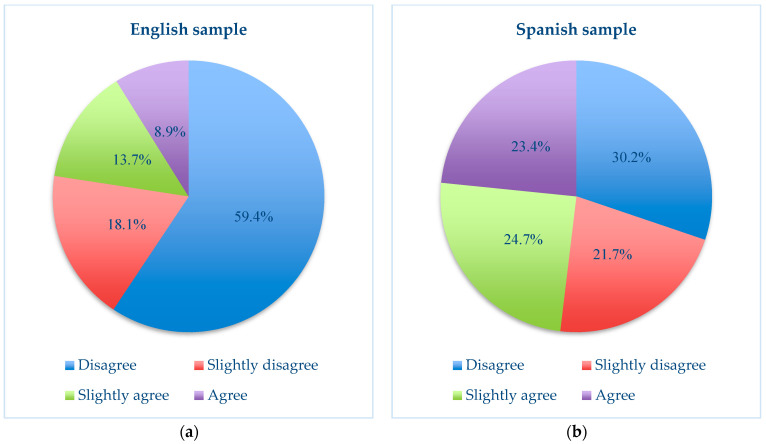
(**a**) Comparison of results for item 6, “Looking at partner’s daily internet connections”, in the English study and (**b**) comparison of results for item 6, “Looking at partner’s daily internet connections”, in the Spanish study.

**Figure 2 ijerph-21-00926-f002:**
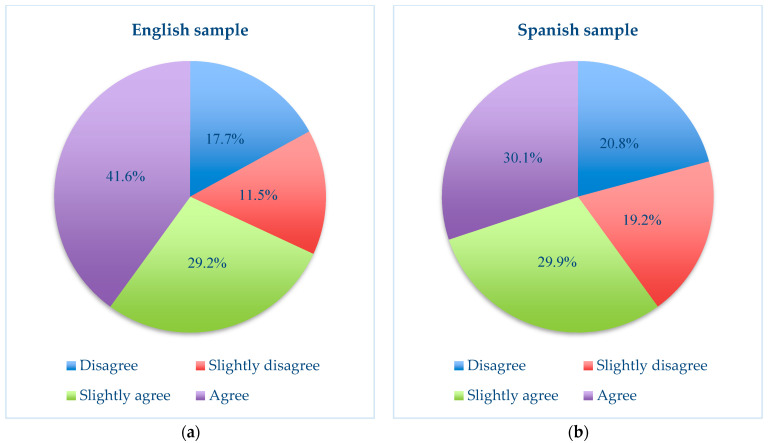
(**a**) Comparison of the results of item 7, “Being constantly in contact through electronic means (mobile, computer) with the partner to know where he/she is or with whom” in the English study and (**b**) comparison of the results of item 7 “Being constantly in contact through electronic means (mobile, computer) with the partner to know where he/she is or with whom” in the Spanish study.

**Figure 3 ijerph-21-00926-f003:**
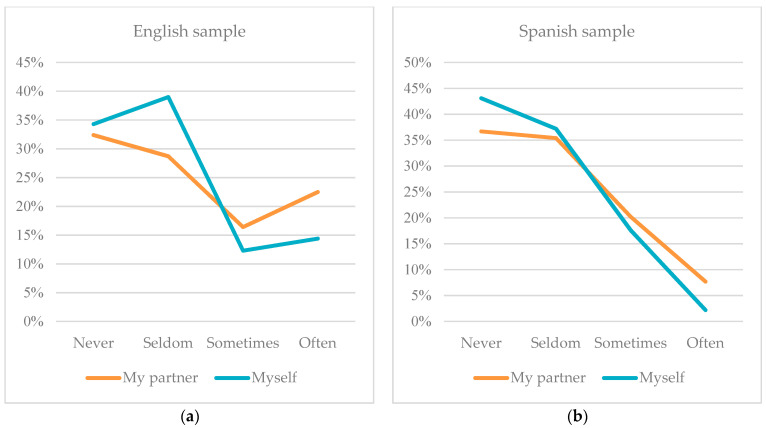
(**a**) Comparison of the results of item 13 “Insistently asking “who are you talking to” in the English study and (**b**) comparison of the results of item 13 “Insistently asking “who are you talking to” in the Spanish study.

**Figure 4 ijerph-21-00926-f004:**
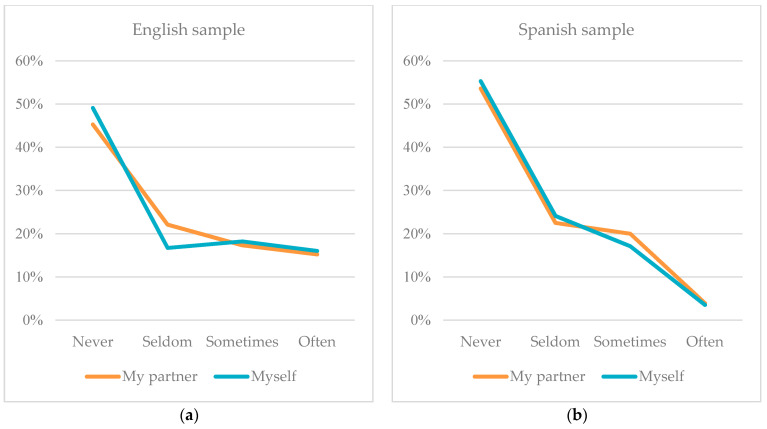
(**a**) Comparison of results for item 34 “Sending scantily clad or nude erotic photos to a partner” in the English study and (**b**) comparison of results for item 34 “Sending scantily clad or nude erotic photos to a partner” in the Spanish study.

**Figure 5 ijerph-21-00926-f005:**
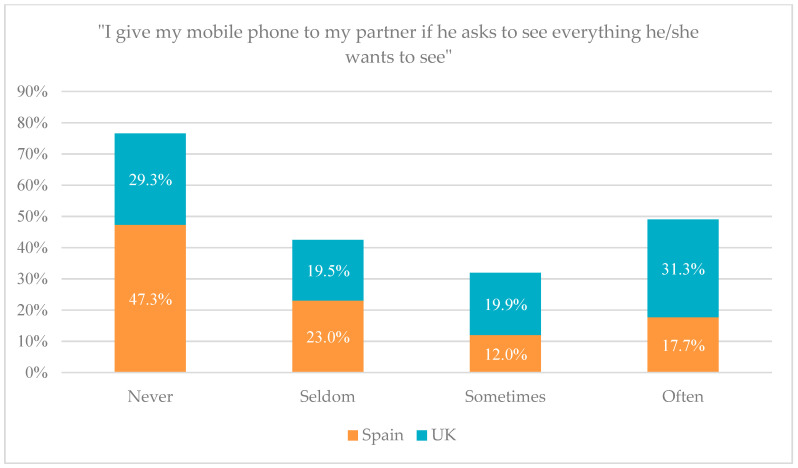
Comparison of results for item 71 on the English and Spanish study.

**Figure 6 ijerph-21-00926-f006:**
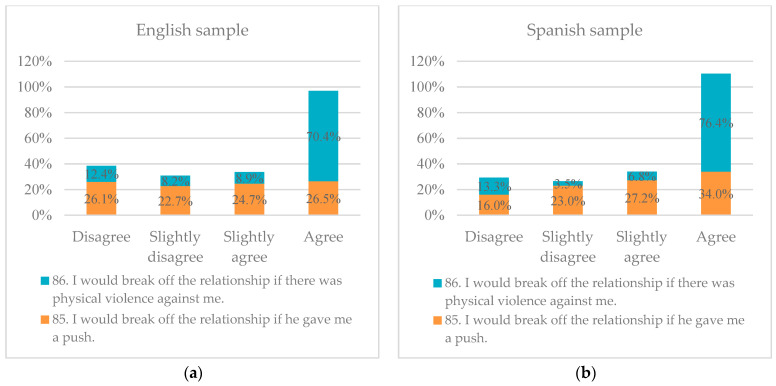
(**a**) Comparison of results on tolerance and possible consequences of physical violence in the English study and (**b**) Comparison of results on tolerance and possible consequences of physical violence in the Spanish study.

## Data Availability

The data presented in this study are available on request from the corresponding author. The data are not publicly available due to its nature.

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
