# Peer review of "Digital Violence in University Student Couples: England vs. Spain"

_ijerph, 2024, doi:10.3390/ijerph21070926_

Round 1
Reviewer 1 Report
Comments and Suggestions for Authors
The title should more clearly indicate the type population being studied.
suggested change:
Digital violence in university student couples: United Kingdom vs SpanThe title should more clearly indicate the type population being studied.
suggested change:
Digital violence in university student couples: United Kingdom vs Span
Author Response
We accept the new title: Digital violence in university student couples: United Kingdom vs Spain
Reviewer 2 Report
Comments and Suggestions for Authors
More information is required regarding the difference between university contexts.
They should explain why they say that the university context may have influenced the results.
The authors include the limitations of the work in the conclusions and it seems to me that this should be included in the discussion.
Compared with other published material it adds a description of the perception that university couples have about digital communication and confirmation of the relevance of the problem.
Authors probably should consider to perform a different statistical analysis to be able to provide significant data to answer the main research question. The conclusions do not exactly respond to the question raised regarding the comparison between the UK and Spain on digital violence in digital communication between university couples. Furthermore, in the conclusions he comments on limitations of the study, which should be commented on in the discussion.
The references are appropriate.
The tables are correct.
Author Response
More information is required regarding the difference between university contexts.
|
Between lines 93 and 98, the following text has been added:
This work is carried out in different university contexts, United Kingdom and Spain. Countries in which, on the one hand, there is a certain similarity in terms of the political and economic situation of the countries, safeguarding cultural and contextual differences; and on the other hand, there has been greater social awareness-raising work in recent years in Spain, especially within the student movement and its university context. |
They should explain why they say that the university context may have influenced the results.
|
There are two main reasons why we are targeting the university population and these have been added to the introduction section:
- University students represent the main recipients of ICTs and a digitally native popu-lation group, because of the age group (lines 42 – 43), in order to analyse how it influences the presence of abusive behaviours. - The nálisis of the university context provides a more easily comparable sample population in terms of the profile of the participants as students. (lines 100 – 102) |
The authors include the limitations of the work in the conclusions and it seems to me that this should be included in the nálisis.
|
We have moved the limitation paragraph to the analysis section. |
The main research question is not clearly explained. It seems that it may be comparing the nálisis the UK and Spain in terms of violence in the digital communication of university couples. For me, it is a question that is not sufficiently documented and it is not sufficiently justified why it is necessary to compare these two contexts. I consider that the nálisis relevant in the field but is currently well studied, but as I said previously, in the case of the study of university couples, it does not sufficiently justify why the university nálisis the UK and Spain.
|
We have reworded the main objective (lines 98 – 99) to clarify it: - the main objective of this research is to analyse the digital violence that occurs through the use of ICT and all commonly used electronic devices in affective-sexual relationships in the university population.
Also the sentence relating to the specific objective of the research, between lines 102 – 104, has been reworded to improve comprehension: - the specific objective of this work is to detect and measure the prevalence of digital violence among university couples, comparing the populations of Spain and the UK, in order to pursue its prevention. |
Compared with other published material it adds a description of the perception that university couples have about digital communication and confirmation of the relevance of the nálisis.
|
Prior to the analysis of the prevalence of digital violence in both contexts, we considered it meaningful to ask about the participants’ perception of this violence in order to contrast the results between beliefs and their own actions. |
Authors probably should consider to perform a different statistical nálisis to be able to provide significant data to answer the main research question. The conclusions do not exactly respond to the question raised regarding the comparison between the UK and Spain on digital violence in digital communication between university couples. Furthermore, in the conclusions he comments on limitations of the study, which should be commented on in the nálisis.
|
We have considered adding more information related to statistical analysis in lines 388 – 394:
Last but not least, item analysis and scale reliability was estimated using Cronbach's alpha coefficient, with a value of 0.955 in the English study and 0.945 in the Spanish study, determining a high internal consistency in the Ad hoc questionnaire. It is also mentioned that the sample distribution did not meet the assumption of normality (p=0.000), calculated through the Kolmogorov-Smirnov test, so non-parametric statistics were used. In addition, KMO (0.872) and Bartlett's sphericity (1.4197.470; p < 0.000) tests were performed to verify the appropriateness of conducting the factor analysis. |
Reviewer 3 Report
Comments and Suggestions for Authors
Thank you for the opportunity to review this manuscript. The manuscript “UK vs. Spain in digital violence in young couples” presents a well conducted study that investigates and studies the violence that occurs through the use of commonly used electronic devices in affective-sexual relationships, in the university population. Considering the broader context of presented studies and further impact of the published papers, I have few corrections /suggestions, suggested to be addressed before next steps in the publication.
1. How were these two universities selected/included in the study? I would see a potential issue of selection bias in selecting and comparing these two cities; and also, universities. The population in Huelva could be only 15-20% that of Lancashire? Also, University of Huelva would enroll only1/3rd of the students that of university of central Lancashire city. The trends would be different of small cities than metropolitan cities?
2. As exclusion criteria, participants with a dating relationship of less than three months were not selected? As per literature the abuse (if any) will be more prominent in the start in conditions where that is not up to the mark? Or the authors have thoughts about other ideas?
3. I would suggest also performing further stratified analysis (if possible) based on age, educational level and possibly familial history/background.
4. The discussion is very well written, however, very short. The readers will find it useful to read about certain results discussion particularly from the authors who have very well analyzed the results.
Minor comments:
The key word “digital violence” is already found in the title and would suggest to replace by a suitable similar word.
The authors have used both the words “educational” and “university” in the last part of the abstract. Was it used in similar context or for different settings?
Author Response
How were these two universities selected/included in the study? I would see a potential issue of selection bias in selecting and comparing these two cities; and also, universities. The population in Huelva could be only 15-20% that of Lancashire? Also, University of Huelva would enroll only1/3rd of the students that of university of central Lancashire city. The trends would be different of small cities than metropolitan cities? |
We selected the University of Central Lancashire in Preston, England, because of the similarities of the context of this city compared to the city of Huelva. The former had a population of 141,818 inhabitants in 2018, the latter with 144,258 inhabitants in the same year. In addition, both cities share similar characteristics in terms of size, development and facilities. In terms of university contexts, both universities enrol international students every year, despite being located in small cities. However, they differ in the university population. |
As exclusion criteria, participants with a dating relationship of less than three months were not selected? As per literature the abuse (if any) will be more prominent in the start in conditions where that is not up to the mark? Or the authors have thoughts about other ideas? |
We focused on affective-sexual relationships established between students and determined that a romantic relationship is longer than three months to be considered. Although, according to the literature, it is true that abuse (if any) may start at the beginning of the relationship, many studies show that these abusive behaviours start to be detected by partners later than three months into the relationship, when the relationship is consolidated. |
I would suggest also performing further stratified analysis (if possible) based on age, educational level and possibly familial history/background. |
The suggested socio-demographic variables were not as explicit in the article because they did not contain significant references in both studies. Nevertheless, we added a few sentences to the text of the results referencing them lLines 176 -188):
(…) followed by 20.8% in a single-parent family in the first study and 12% in the second. Regarding the socio-economic status of the family, the majority of respondents in both samples said they were in the middle socio-economic status, 47.3% and 48.6% said they were in the upper-middle range, while another 49.8% and 45.1% said they were in the lower-middle, in Spain and the UK respectively. Looking at the educational level of the family, both studies show very similar results again. The mother figure obtained a majority of 67.8% with "General Certificate of Secondary Education (GCSE)" in the English study, while in the Spanish study the percentage was 59.9%, both followed by the option of "Educated up to Primary Education level". The parent figure was 74.6% and 18.2% for "General Certificate of Secondary Education (GCSE)" and "Certificate of Higher Education or more" in the English questionnaire; 78.3% and 14.7% for "General Certificate of Secondary Education (GCSE)" and "Certificate of Higher Education or more" in the Spanish questionnaire. |
The discussion is very well written, however, very short. The readers will find it useful to read about certain results discussion particularly from the authors who have very well analyzed the results. |
We are very grateful for the appreciation of the discussion. In addition, we consider adding the following sentences (lines 433 – 436) and the limitations of the research (lines 441 – 451):
The results show that digital violence can be perpetrated without the presence of physical violence. However, cases in which physically and sexually violent behaviour is detected are also likely to show abuse, harassment and control through electronic devices. |
|
|
Minor comments: |
|
The key word “digital violence” is already found in the title and would suggest to replace by a suitable similar word. |
We repeat "digital violence" as a key word due to the lack of consensus in the conceptualisation of this phenomenon. Therefore, we reiterate and describe the term "digital violence" as any action of control, invasion of privacy, manipulation of personal information, insistent and constant communication with the partner, coercion, emotional abuse (blackmail, criticism, accusations, isolation or blocking in social networks, etc.), threats and public or private humiliation through electronic devices. |
The authors have used both the words “educational” and “university” in the last part of the abstract. Was it used in similar context or for different settings? |
It was used in a similar context. However, "educational" has been removed from the text because it might sound redundant and to avoid possible confusion. |
Round 2
Reviewer 2 Report
Comments and Suggestions for Authors
This manuscript could already be published
Author Response
We appreciate your review and comments that have helped to substantially improve our paper.
Greetings
Reviewer 3 Report
Comments and Suggestions for Authors
Thank you for the opportunity to review the revised version of the manuscript. The manuscript “Digital violence in university student couples: United Kingdom vs Spain” presents a well conducted study that investigates and studies the violence that occurs through the use of commonly used electronic devices in affective-sexual relationships, in the university population. The manuscript has improved since the previous version. I have no further comments.
Author Response

(The authors gave the same response as above.)
